# Relationship between Dairy Cow Health and Intensity of Greenhouse Gas Emissions

**DOI:** 10.3390/ani14060829

**Published:** 2024-03-07

**Authors:** Karina Džermeikaitė, Justina Krištolaitytė, Ramūnas Antanaitis

**Affiliations:** Large Animal Clinic, Veterinary Academy, Lithuania University of Health Sciences, Tilžės Str. 18, LT-47181 Kaunas, Lithuania; justina.kristolaityte@stud.lsmu.lt (J.K.); ramunas.antanaitis@lsmu.lt (R.A.)

**Keywords:** global warming, greenhouse gas, methane emissions, cattle, cattle health

## Abstract

**Simple Summary:**

There is increasing pressure on dairy systems worldwide to reduce greenhouse gas (GHG) emissions. Anticipated improvements in animal welfare and reduced mortality and disease rates are expected to result in increased animal productivity, hence reducing GHG emissions per unit of output. There has been a lack of focus on quantifying the intensity of GHG emissions and their impact on the health of dairy cows, encompassing both subclinical and clinical problems. The initial indications of illnesses in dairy cows typically manifest as a decline in both feed consumption and milk production. The implementation of emission-reducing strategies and the maintenance of optimal animal welfare are suggested as effective approaches to improve the sustainability of modern dairy production. More information regarding animal health prior to animal mortality is required to enhance the precision of the estimate for GHG emission intensity. Through the analysis of animal health data prior to mortality, a more accurate estimation of GHG emission intensity can be attained. This will allow dairy producers to have a better understanding of the environmental consequences of their operations and make well-informed choices regarding ways to reduce emissions. In addition, placing a high priority on achieving the best possible conditions for animal welfare not only supports the long-term viability of dairy production, but also guarantees the health and happiness of the cows. This, in turn, can result in enhanced overall productivity and a decrease in the occurrence of diseases. Therefore, it is essential to adopt a comprehensive approach that considers both environmental and animal health aspects to advance a sustainable and efficient dairy sector. Furthermore, the integration of technology such as precision farming and data analytics can significantly augment the efficiency and sustainability of dairy operations.

**Abstract:**

The dairy industry is facing criticism for its role in exacerbating global GHG emissions, as climate change becomes an increasingly pressing issue. These emissions mostly originate from methane (CH_4_), nitrous oxide (N_2_O), and carbon dioxide (CO_2_). An optimal strategy involves the creation of an economical monitoring device to evaluate methane emissions from dairy animals. Livestock production systems encounter difficulties because of escalating food demand and environmental concerns. Enhancing animal productivity via nutrition, feeding management, reproduction, or genetics can result in a decrease in CH_4_ emissions per unit of meat or milk. This CH_4_ unit approach allows for a more accurate comparison of emissions across different animal production systems, considering variations in productivity. Expressing methane emissions per unit allows for easier comparison between different sources of emissions. Expressing emissions per unit (e.g., per cow) highlights the relative impact of these sources on the environment. By quantifying emissions on a per unit basis, it becomes easier to identify high-emission sources and target mitigation efforts accordingly. Many environmental policies and regulations focus on reducing emissions per unit of activity or output. By focusing on emissions per unit, policymakers and producers can work together to implement practices that lower emissions without sacrificing productivity. Expressing methane emissions in this way aligns with policy goals aimed at curbing overall greenhouse gas emissions. While it is true that total emissions affect the atmosphere globally, breaking down emissions per unit helps to understand the specific contributions of different activities and sectors to overall greenhouse gas emissions. Tackling cattle health issues can increase productivity, reduce GHG emissions, and improve animal welfare. Addressing livestock health issues can also provide favourable impacts on human health by reducing the prevalence of infectious illnesses in livestock, thereby mitigating the likelihood of zoonotic infections transmitting to humans. The progress in animal health offers the potential for a future in which the likelihood of animal diseases is reduced because of improved immunity, more effective preventative techniques, earlier identification, and innovative treatments. The primary objective of veterinary medicine is to eradicate clinical infectious diseases in small groups of animals. However, as the animal population grows, the emphasis shifts towards proactive treatment to tackle subclinical diseases and enhance production. Proactive treatment encompasses the consistent monitoring and implementation of preventive measures, such as vaccination and adherence to appropriate nutrition. Through the implementation of these measures, the livestock industry may enhance both animal well-being and mitigate the release of methane and nitrous oxide, thereby fostering environmental sustainability. In addition, advocating for sustainable farming methods and providing farmers with education on the significance of mitigating GHG emissions can bolster the industry’s endeavours to tackle climate change and infectious illnesses. This will result in a more robust and environmentally sustainable agriculture industry. This review seeks to conduct a thorough examination of the correlation between the health condition of cattle, the composition of milk produced, and the emissions of methane gas. It aims to identify areas where research is lacking and to provide guidance for future scientific investigations, policy making, and industry practices. The goal is to address the difficulties associated with methane emissions in the cattle industry. The primary global health challenge is to identify the causative relationship between climate change and infectious illnesses. Reducing CH_4_ and N_2_O emissions from digestive fermentation and animal manure can be achieved by improving animal well-being and limiting disease and mortality.

## 1. Introduction

Climate change is the gradual alteration of temperature and weather patterns caused by the accumulation of heat-trapping gases in the atmosphere over an extended period. The Bulletin of the American Meteorological Society’s global climate research reveals that the seven hottest years since the mid to late 1800s took place between 2015 and 2021 [1,2]. In the last ten years, there has been a growing level of consciousness regarding climate change resulting from the rise in GHG emissions [3,4]. The dairy industry is facing growing criticism for its contribution to global GHG emissions [5]. As a result, there has been an extraordinary sustainable intensification in advanced dairy farming worldwide in order to produce milk more efficiently [6]. Globally, ambitious goals have been established to decrease GHG emissions. At the recent UNFCCC 26th Conference in Glasgow, over 120 countries made a solemn commitment to achieve net zero emissions by 2050–2070 [7]. The primary sources of GHG emissions from ruminants are methane (CH_4_), nitrous oxide (N_2_O), and carbon dioxide (CO_2_) [8]. Enteric fermentation, feed production, manure production, and management operations are the main sources of these emissions [9]. Dairy production contributes to global warming because of the emission of methane, a highly potent GHG [10]. Methane is created in the rumen during the regular fermentation process by methanogenic archaea using either CO_2_ and hydrogen (H_2_), methylamines or methanol, or acetate and H_2_ to produce CH_4_ [11]. CH_4_ is a gas with a greenhouse potential of 25 times that of CO_2_ [12].

The scientific community regards global warming as a substantial issue [13]. It is widely accepted that human activities, particularly the burning of fossil fuels, are the main cause of the increase in greenhouse gases in the atmosphere. This has resulted in a rise in global temperatures, leading to numerous environmental consequences such as melting ice caps, rising sea levels, and extreme weather events. Numerous studies and data support the scientific consensus that there is an urgent need to address global warming, so it is crucial for governments, organisations, and individuals to take immediate action to mitigate its effects [14].

Ruminant animals, such as beef and dairy cattle, are significant methane producers due to enteric fermentation that takes place in their rumen during the digesting process [11,15]. Research reveals that cattle, which comprise both meat and milk production, produce around 3.8 gigatons of carbon dioxide equivalent each year, accounting for 62 percent of the overall emissions from livestock. Pigs are responsible for 14 percent of the emissions, chickens for 9 percent, buffaloes for 8 percent, and small ruminants for 7 percent (Figure 1) [16].

Presently, the annual global methane emissions amount to around 500–600 tera-gram. Methanogenesis from diverse ecosystems accounts for over 70% of these emissions [17]. This is the last stage of the process of breaking down organic matter without the presence of oxygen, after all the inorganic substances that accept electrons, such as nitrate, ferric iron, or sulphate, have been used up [18]. Acetolactic methanogens decompose acetate into methane and carbon dioxide. They inhabit environments where hydrogenotrophic methanogens decrease hydrogen gas (H_2_) levels to a point where optimal circumstances for abundant acetate production are created. Acetolactic methanogens are primary methane producers in anaerobic digesters, rice fields, and wetlands, responsible for two-thirds of biologically produced methane emissions. They decompose acetate into methane and carbon dioxide, thriving in environments where hydrogenotrophic methanogens decrease H_2_ levels. Hydrogenotrophic methanogens utilise H_2_, formate, or a small number of simple alcohols as energy sources. They then convert CO_2_ into CH_4_ through reduction and are the main producers of methane in deep marine sediments, termite hindguts, and the gastrointestinal tracts of humans and animals [17,18]. Methanogens are the primary producers of methane in deep sea sediments, termite hindguts, and the gastrointestinal systems of humans and animals. Collectively, these sources account for one-third of the methane emissions produced by living organisms (Figure 2) [17].

Approximately 90% of the CH_4_ that dairy cows produced comes from their breath and ejected rumen gases; CH_4_ is also a loss in energy and the result of fermentation by methanogens like archaea [12]. Therefore, a desirable approach would be to develop a straightforward, resilient, and cost-effective monitoring technology that can be widely implemented to assess CH_4_ emissions from dairy animals [19].

Livestock production systems face challenges posed by increasing food demand and environmental issues. When animal productivity is improved through nutrition, feeding management, reproduction, or genetics, CH_4_ production per unit of meat or milk is reduced. A 20% reduction in total CH_4_ production could allow growing cattle to gain an additional 75 g/d of body weight and 1 L/d more milk yield (MY) from dairy cows [20]. The rise in animal productivity led to a decrease in enteric CH_4_ emissions per unit of animal production (milk and average daily gain) and an enhancement in feed efficiency [21]. According to research, there is a heritability of enteric CH_4_ production and a genetic correlation with the intake of milk lactose, protein, fat, and DM [22]. Milk production and lifetime performance play a significant role in the breeding of high-production dairy cattle like Holstein cows. Enteric methane production is strongly associated with the genetics, health, and productivity of dairy cows, as well as with feeding and nutrition management [23]. On a global scale, it is estimated that livestock diseases result in a 25% decrease in productivity [24]. Infectious disorders can worsen these contributions by increasing methane emissions linked to animal production (Figure 3). The rise in the prevalence of numerous contagious illnesses has led to the development of a dangerous cycle involving climate, livestock, and disease, which poses a significant and imminent danger [7]. Improving the health and reproductive state of a herd could help by reducing the number of animals that have to be culled against their will and increasing fertility traits like calving intervals. This would cut down on unnecessary costs and CH_4_ production [25]. Therefore, addressing livestock health issues may boost output while also lowering the intensity of GHG emissions and enhancing animal welfare [24].

A worldwide goal is to lessen the damage that animal production does to the environment. Innovative ideas and tools can help the change to a more environmentally friendly livestock system [26]. Additionally, addressing livestock health issues can also have positive effects on human health. By reducing the prevalence of contagious illnesses in livestock, the risk in zoonotic diseases spreading to humans can be minimised. This would not only protect human populations from potential outbreaks but also reduce the burden on healthcare systems and resources. Overall, prioritising livestock health can have far-reaching benefits for both animal welfare and public health.

The advancements in animal health present the possibility of a future where the risk of animal diseases is significantly diminished. This is due to enhanced immunity, better prevention methods, earlier and more precise detection, and novel therapies. Maintaining the well-being of animals not only mitigates pollutants stemming from livestock production, but also diminishes the likelihood of spreading infections to people [27].

This review aims to provide a comprehensive analysis of the current understanding of the relationship between health status, milk composition, and methane emissions in cattle. By identifying the gaps in research and highlighting areas of deficiency, this review will contribute to the development of effective strategies for mitigating methane emissions while ensuring the well-being of cattle. Additionally, by exploring potential future outlooks, this review will help guide future scientific studies, policy decisions, and industry practices to address the challenges associated with methane emissions in the cattle sector.

## 2. Dairy Cow Health and Methane Emissions

Enteric fermentation, the process of food digestion in ruminant animals like cattle and sheep, results in the production and release of methane [15]. CH_4_ is primarily expelled or eructated via the nasal and oral cavities as a by-product of anaerobic fermentation in the rumen, a process in which ruminal microorganisms convert feed into nutrients that are readily absorbable by the host animal, with 97% being expelled by the mouth and 3% through the rectum [28,29,30]. Methanogens are primarily responsible for producing CH_4_ during the anaerobic degradation of plant biomass in the rumen (Figure 4) [29]. The molecular process of methane production in the gastrointestinal tract is extensively understood. Elevated fibre intake leads to an increase in rumen pH and a reduction in the rate at which digesta moves through the gastrointestinal tract. Consequently, a change in ruminal fermentation towards acetate leads to an increase in the amount of dissolved hydrogen equivalents available to produce CH_4_ by rumen methanogens [31].

Microbial fermentation is a complex series of steps that begins with the breakdown of dietary polysaccharides into easily digestible sugars through the action of microbial enzymes. Additionally, after a series of multi-step processes, these sugars that can undergo hydrolysis go through fermentation to produce volatile fatty acids (VFAs), frequently acetate, propionate, and butyrate, without the presence of oxygen [32]. These VFAs serve as an energy source [33]. Throughout the process, elemental hydrogen (H), which acts as a reducing agent and is commonly referred to as metabolic hydrogen, is generated as a secondary product. Hydrogen-producing bacterial species convert the metabolic hydrogen into molecular H_2_, which methanogens then convert into CH_4_ [32].

Animal health is fundamental to the establishment of a sustainable animal agriculture industry [34]. The major global health challenge lies in establishing the correlations between climate change and infectious disease [35]. Enhancing animal well-being and minimising animal sickness and death to improve the effectiveness of the animal production system present possibilities for decreasing both CH_4_ and N_2_O emissions resulting from digestive fermentation and animal manure [24,36].

The primary objective of veterinary medicine in livestock production systems that depend on small herds is the complete elimination of clinical infectious illnesses, with a particular emphasis on treating each animal individually. Nevertheless, when the number of animals in a herd and their output rise, the attention turns to proactive veterinary treatment and places more importance on addressing subclinical diseases and implementing comprehensive health management programmes that aim to enhance productivity. Irrespective of the stage of development of a livestock production system, a decrease in the number of deaths and illnesses results in a higher amount of products that can be sold, which in turn, reduces GHG emissions per unit of product [36]. Animals typically respond to diseases by initially decreasing their consumption of food, which subsequently leads to decreased productivity and increased GHG emissions per unit of output [37]. The primary factors influencing the fluctuation in methane emissions from ruminant animals, including cattle, buffalo, sheep, goats, and camels, are the intake and quality of their feed. Increased consumption of feed and/or decreased quality of feed results in higher methane emission. Typically, larger animals exhibit higher feed intake demands [38].

Furthermore, the process of calving induces significant stress in cows, with around 65% of all disease occurrences in the dairy herd taking place during this period [39]. Additional research indicates that around 75% of infections in dairy cows tend to arise within the initial month following calving. The periparturient phase, also known as the transition period, typically spans from 3 weeks before giving birth to 3 weeks after giving birth [40]. During this stage, cows experience a condition known as negative energy balance. This occurs when the demand for nutrients for milk production increases rapidly and surpasses the supply of nutrients obtained from food consumption [41].

The epidemiology of infectious animal diseases can be significantly influenced by climate change, which is directly connected to production environments and their subsequent consequences [42]. The increase in GHG emissions per ton of milk in ill cows compared to healthy cows might reach up to 25%, depending on the specific health disorder. The estimated increases in GHG emissions per unit of milk and per case are 7%, 8%, and 16% for mastitis, lameness, and infertility, respectively [8]. The first table displays the data extracted from the literature regarding the economic consequences of specific diseases, the variations in milk production, and the influence on GHG emissions (Table 1).

Only a limited number of studies have investigated the relationship between dairy cow health and its impact on GHG emissions [37]. Ensuring the future viability of the dairy business necessitates the careful management of dairy cow health in conjunction with environmental sustainability. This requires adopting a comprehensive approach that considers both the well-being of the animals and the ecological consequences. The economic viability of livestock is significantly affected by the repercussions of livestock disease. However, there is a lack of comprehensive and up-to-date literature on the economic and environmental consequences of cattle diseases, which is a matter of worry for producers [43].

### 2.1. Metabolic Diseases

During the transition from late gestation to early lactation, metabolic and hormonal profile changes occur. Homeorhetic modifications take place during this phase to supply nutrients to the neonate and facilitate lactogenesis. The alterations linked to the heightened nutritional requirements and reduced food consumption that transpire throughout this stage contribute to the formation of an adverse energy balance (NEB). This metabolic and nutritional imbalance has the potential to contribute to the development of an immunosuppressive state [56]. An energy deficit causes a decrease in blood glucose levels and prompts the body to use its stored reserves for more energy. This leads to higher levels of non-esterified fatty acids (NEFA) and β-hydroxybutyric acid (BHBA) in the blood [41]. These parameters are generated as metabolites during the lipid oxidation of fatty acids in the liver. The liver in ruminants maintains energy balance by converting propionic acid, which is taken from the rumen, into glucose. Additionally, it controls fat metabolism by both oxidising and synthesising fat. The liver plays a crucial role in regulating the body’s metabolism and maintaining energy equilibrium. Moreover, the liver’s metabolic and pathological states have a significant impact on animal productivity. Due to its strong association with feed efficiency and energy level, liver metabolism is thought to play a direct and indirect role in intestinal methane generation [57]. Understanding and optimising liver metabolism in ruminants is, therefore, important for reducing methane emissions and improving overall animal productivity.

Additionally, calcium and phosphorus are released to produce milk, resulting in a reduction in their levels in the bloodstream. These metabolic alterations can result in hypocalcaemia, ketosis, displaced abomasum, and hepatic lipidosis [41]. These illnesses are associated with reduced milk production, decreased conception rates, longer intervals between calving, lameness, and diminished well-being and productive lifetime in the herd [58].

Ketosis is a pronounced metabolic disorder that manifests in dairy cows during the initial stages of lactation. It is defined by increased concentrations of ketone bodies in the bloodstream, which frequently result in decreased efficiency, reproductive problems, and occasionally, mortality or culling [59]. For instance, a study demonstrated that by decreasing subclinical ketosis (SCK) and associated illnesses in dairy cows, it is possible to lower GHG emissions per unit of milk produced. The mean rise in GHG emissions per unit of SCK was 20.9 kg of carbon dioxide equivalent per metric ton of fat-and-protein-corrected milk (CO_2_e/t FPCM), representing a 2.3% increase [51]. Hence, the occurrence of ketosis in a dairy cow herd may contribute to the intensity of GHG emissions.

Sub-acute ruminal acidosis (SARA) is a widely acknowledged digestive condition that affects high-producing dairy cows. It has detrimental effects on both the health of the animals and the profitability of the herd, especially in well-managed dairy farms [55]. SARA in cattle can cause physiological disruptions resulting from decreased dry matter intake (DMI), which may lead to acute ruminal acidosis. Diagnosing SARA at an individual level is challenging due to the vague and frequently observable clinical signs mostly seen at the herd level. Nevertheless, it can be recognised by a decrease in ruminal pH [60,61]. SARA typically arises when the pH level in the rumen remains between 5.2 and 6 for an extended duration [55]. Using real-time measured reticulorumen parameters, Antanaitis et al. [62] discovered that dairy cows whose reticulorumen pH ranged from 6.22 to 6.42 had an average total methane emission increase of 46.18% [62]. The data demonstrate that ruminal pH has a significant impact on the physiology and fermentation of the rumen, which in turn, affects methanogenesis [63]. On the other hand, several studies have shown that a decrease in pH promotes the synthesis of propionate, providing alternative routes for the elimination of hydrogen (H) ions. This leads to a decrease in the amount of hydrogen accessible for butyrate, which affects fibrolytic bacteria and methanogens, thus resulting in a reduction in CH_4_ synthesis [64,65,66]. Moreover, it may be linked to a decrease in the diversity of methanogens and changes in composition such as *Methanobrevibacter* spp. and *Methanosphera* spp. that were observed in SARA [63]. The bacterial population underwent alterations in response to acidosis in the laboratory investigation of M. Eger et al. [67], and it was restored to its original state 5 days following the acidosis challenge [67]. Considering that fibrolytic and amylolytic bacteria, as well as lactobacilli, have different pH preferences, the drop in pH may reflect most of the changes that were pointed out. Unlike bacteria, the population of methane-producing archaea was unable to be restored after acidosis. It is crucial to emphasise that the decrease in rumen pH can only be used to identify the ideal pH ranges that are advantageous for developing methods to reduce methane emissions, such as using feeding management techniques and supplementing with prebiotics and probiotics while not jeopardising the animal’s well-being [68].

### 2.2. Mastitis

Ensuring optimal udder health is crucial for both dairy farmers and the entire dairy production chain to provide milk of superior quality. Nevertheless, the production industry faces ongoing challenges from infections, with mastitis being a disease that has significant economic implications. Mastitis is an inflammatory condition of the mammary gland that occurs in dairy cows around the time of calving. It is categorised as one of the periparturient diseases [69]. Clinical mastitis (CM) is an infection that occurs inside the mammary gland and leads to a decrease in milk production and fertility. It also increases the rate at which cows are removed from the herd and the rate of cow deaths. As a result, CM has a detrimental effect on the efficiency of milk production, which is measured by the ratio of output to input. This could potentially lead to an increase in GHG emissions per unit of product [70].

Monitoring mastitis in cows has become a regular practice since the 1980s, with monthly somatic cell count (SCC) examinations being conducted. Even in the present day, this strategy continues to be widely acknowledged and embraced [71]. Özkan Gülzari et al. [72] showed that by lowering the SCC in milk production from 800,000 cells/mL to 50,000 cells/mL, it is possible to decrease the overall emissions intensity of farms by 3.7% [72]. Subclinical mastitis, a variant of the disease characterised by the absence of obvious indications of infection in the udders, diminishes both milk production and feed consumption in cows [73]. The anticipated disparity in DMI between a cow with a higher SCC of 250,000 cells/mL and a cow with a lower SCC of 50,000 cells/mL would result in an additional 12 grammes of methane per day (equivalent to 4.4 kg of methane per year). This accounts for approximately 2.8% of the annual enteric methane emissions from a dairy cow in the United States in 2017. Additionally, there would be an extra 0.34 g of methane per kilogramme of milk, representing a 2.8% increase in methane emissions per kilogramme of milk produced [74,75]. According to a recent study, subclinical mastitis can cause a significant increase in both enteric and manure methane emissions [73]. Specifically, infected cows can produce up to 8% more methane per kilogramme of milk compared to healthy cows [72].

Therefore, research indicates that clinical mastitis has a detrimental effect on output, namely reducing feed efficiency (measured as kg of milk per kg of feed intake) in cows. Consequently, this leads to an increase in GHG emissions per unit of product produced. The scientific findings indicate that cows with clinical mastitis have, on average, a 57.5 (6.2%) kg CO_2_e/t FPCM higher emission compared to cows without clinical mastitis [70]. It is necessary to emphasise that a rise in temperature of 10 °C in different regions of the United States resulted in an increase in antibiotic resistance of 4.2%, 2.2%, and 2.7% for the common pathogens *Escherichia coli*, *Klebsiella pneumoniae*, and *Staphylococcus aureus*, respectively. This demonstrates the intricate and mutually influential relationship linking animals, disease, and climate [76]. Implementing measures to prevent clinical mastitis can serve as a viable approach for farmers to mitigate GHG emissions and promote the sustainable growth of the dairy industry. This practice can also enhance farmers’ revenue and improve the well-being of cows [70].

### 2.3. Lameness

Lameness is a principal issue in terms of health and well-being in dairy farming [47]. Numerous factors contribute to this painful condition, which has detrimental economic effects such as decreased production, decreased fertility, and an increased likelihood of culling [77]. When productivity goes down, environmental effects get worse per unit of production because each animal’s maintenance costs rise in relation to production, so more animals are needed to keep milk production steady [78]. As herd sizes expand, lameness is thought to become more widespread and severe [79]. Approximately 90 percent of instances of lameness are linked to foot lesions [80]. Implementing management measures aimed at decreasing foot lesions might potentially lead to a decrease in GHG emissions per kilogram of milk produced [81]. The different increase in global warming potential (GWP) with a time horizon of 100 years (kg CO_2_-eq) due to lameness indicated that higher prevalence of foot diseases has a more negative impact. The result suggests that implementing sensors as well as information and communication technology for lameness detection could enhance management on dairy farms. This, in turn, could reduce the negative environmental effects linked to lameness by addressing the increased cow–handler ratio caused by larger herd sizes [79]. A further study discovered that foot lesions led to a mean rise of 13.6 (1.5%) kg CO_2_e/t of fat-and-protein-corrected milk (FPCM) in GHG emissions. Moreover, it also varied based on parity, with a rise from 17 kg CO_2_e/t FPCM in parity 1 to 7 kg CO_2_e/t FPCM in parity 5. This was mostly attributed to the heightened influence of eliminating cows with a low parity, since the rearing of a young female bovine results in the generation of GHG without yielding any milk production. Additionally, the emissions varied according on the kind of foot injury. The impact of non-infected white line disease on GHG emissions was the most significant, whereas infectious digital dermatitis had the least impact. Nevertheless, despite its negligible effect, viral digital dermatitis has the highest occurrence rate, and hence made the greatest contribution to the total impact of foot lesions. The findings of this research demonstrate that the assessment of GHG emissions by type of foot lesion provides more valuable information and has the potential to successfully reduce GHG emissions from the dairy sector [81]. This research not only highlights the importance of understanding the diverse types of foot lesions in dairy cows, but also emphasises the need for targeted management strategies to reduce GHG emissions.

### 2.4. Parasites

In livestock production, helminth infections are pervasive and have detrimental effects on feed ingestion, growth, productivity, reproductive performance, welfare, and health status. According to reports, nematode infection ranks second in terms of healthcare expenditures among dairy producers, following mastitis. Additionally, they contribute to the escalation of GHG emissions linked to ruminant agriculture [82]. Managing gastrointestinal parasites has the potential to decrease GHG emissions in grazing livestock. Nevertheless, the impact of these factors on methane emissions remains uncertain due to a dearth of studies [83]. Sheep that were infected with larvae of *Haemonchus contortus* and *Trichostrongylus colubriformis* had greater levels of CH_4_ per unit of consumed DM compared to the uninfected sheep (10.72 vs. 6.75 CH_4_/DMI (g/kg DM), *p*  <  0.05) throughout the assessed time [84]. Repeatedly infecting ewes with *Teladorsagia circumcincta* infective larvae resulted in a 16% increase in total methane yield and a 4% increase in total nitrous oxide yield per unit of dry matter intake. Similarly, per unit of digestible organic matter intake, there was a 46% increase in total methane yield and a 31% increase in total nitrous oxide yield [85]. Just recently, Fox et al. [83] discovered that lambs infected with abomasal parasites exhibited a 33% increase in total CH_4_ output compared to uninfected animals. As per the findings of N. J. Fox et al. [83] and J. G. Houdijk et al. [85], using deworming measures for female sheep may enhance both production efficiency and environmental sustainability in sheep farming. This approach may also be relevant for cattle [83,85].

One of the limited number of studies identified examined the effects of *Fasciola hepatica* in beef cattle. In beef cattle, the observed 1.5% rise in GHG emissions intensity in 2022 attributed to *Fasciola hepatica* seems to be moderate. Nevertheless, the presence of liver fluke also leads to alterations in feed conversion ratio, milk production levels and the quality of output. Therefore, eliminating the fluke challenge would have a significantly larger effect on emissions intensity in real-world scenarios than what is observed in current studies [86]. However, further research should be conducted on dairy cows of this nature.

### 2.5. Viral Infections

Given the significance of viral illnesses in worldwide cow production, it is crucial to make efforts to eliminate, or at least decrease, the occurrence of these diseases. Environmental change, trade globalisation, and livestock expansion have all played a role in the dissemination of established pathogens and the introduction of disease into formerly disease-free regions and animal populations [87]. Contagious viral infections in dairy cattle have significant consequences for milk supply, quality, and general animal well-being [88]. The GHG emissions intensity (kg CO_2_eq/kg product) of livestock-derived food production can be reduced by decreasing the incidence or eradicating diseases that negatively affect milk and meat production. However, the extent of specific disease effects differs based on factors such as output losses, disease prevalence, and baseline population characteristics [89].

Research conducted in the United Kingdom (UK) revealed that bovine viral diarrhea (BVD) might raise greenhouse gas emissions per unit of beef carcass by up to 113% and increase emissions by 14% compared to the healthy baseline for dairy beef production. However, it is important to note that this effect was not adjusted for the prevalence of the illness [90]. In addition, an experiment investigating the impact of dairy bovine illness on GHG emission revealed that eliminating BVD would result in a 4% reduction in GHG emission for average UK herds, while the most problematic 10% of herds would see an 11% decrease [90]. At the herd level, several studies have shown that infected bovine rhinotracheitis results in an 8% increase in GHG emissions per kilogramme of energy-corrected milk and a 20% increase per kilogramme of beef [43]. Another study has shown that by reducing the incidence of foot and mouth disease in beef cattle from 45% to 5%, there would be an 11.3% decrease in GHG emission intensities (CO_2_eq/kg CW) [89]. These findings highlight the significant contribution of diseases in livestock to GHG emission, and the potential for disease control measures to mitigate emissions in the agriculture sector.

Paratuberculosis has long been recognised as a latent issue in dairy cows on a global scale. In the research study conducted by McAloon et al. [91], it was demonstrated that dairy cows that tested positive for a certain condition saw a reduction in MY by 5.9%. There is a lack of data about the emission intensity, nevertheless, it can be inferred that if the nutrients are not well utilised, it would have an impact on the emission intensity [37]. Additionally, further investigation should be undertaken with dairy cows of this sort.

### 2.6. Methane Emissions and Cow Blood Parameters

Monitoring and optimising specific blood parameters in cattle can potentially enhance their overall health, increase efficiency by reducing methane emissions, and improve the growth performance of their offspring, so increasing profitability [92]. Macro- and microminerals are essential for maintaining good cattle production performance by fulfilling the basic physiological needs. Their presence in the bloodstream is crucial for various physiological activities, including health maintenance, growth, reproduction, and the proper functioning of the immunological and endocrine systems [93]. In a recent research investigation, Reintke et al. [92] discovered a correlation between a low blood serum BHB level and a reduction in CH_4_ emissions in ewe. Furthermore, in the Merinoland breed, elevated zinc levels during lactation were linked to decreased methane emissions. The features of interest were not significantly influenced by the serum levels of Na, K, P, glutamate dehydrogenase (GLDH), and Fe [92].

Ľubomíra Grešáková et al. [23] conducted a study that sought to clarify the relationship between the mineral status of dairy cows and enteric methane production at various stages of lactation. Nevertheless, this investigation did not discover any association between exhaled methane emissions and blood plasma mineral status at various stages of breastfeeding [23].

Due to its strong association with feed efficiency and energy level, liver metabolism is assumed to have a direct and indirect role in intestinal methane generation. Scientists conducting experiments on Japanese Black cattle discovered that cattle with high methane emissions (HME) had elevated levels of blood β-hydroxybutyric acid concentration and total ketone bodies compared to cattle with low methane emissions (LME). The rumen-generated butyrate is then transformed into BHBA and delivered via circulation to be used as energy in different tissues. Therefore, the elevated BHBA concentrations detected in the serum of the HME cattle in this investigation may be attributed, at least in part, to the increased rate of butyrate generation in the rumen [57]. The blood metabolites and CH_4_ production is affected by both DMI and diet. L. T. C. Ornelas et al. [94] observed that despite feeding the animals the same diet and them exhibiting identical DMI values during respiration and digestibility assays, the group with low emissions’ CH_4_ production (LPr) exhibited a reduced insulin concentration in comparison to the group with high emissions’ CH_4_ production (HPr). The elevated glucose-to-insulin ratios and reduced insulin levels can be rationalised by variations in DMI during the pre-experimental phase. These distinctions suggest that insulin and the glucose-to-insulin ratio may serve as indirect indicators for animals whose methane production is below one gramme per day [94]. Consistent findings were shown in research conducted by M. Kim et al. [57]. The insulin levels in HME cattle were significantly increased compared to the LME group, indicating that the HME group effectively used amino acids as energy sources in muscle and peripheral tissues to offset the energy depletion caused by methane generation. Insulin facilitated the transportation of amino acids, hence sustaining elevated amounts [57].

## 3. Heritability

Recent studies have demonstrated that dairy cattle have a moderate degree of heritability with regard to CH_4_ characteristics [95,96]. Hence, opting for animals with low CH_4_ emissions is a robust strategy to mitigate CH_4_, since genetic advancements are enduring and accumulate throughout successive generations [96]. There has been a recent surge in interest in choosing animals that have a lower intestinal CH_4_ yield phenotype [97]. While management and food solutions have been thoroughly studied, genetics offers the advantage of having additive and irreversible effects of selection [22]. However, the analysis of the heritability of CH_4_ production is insufficient. This is attributed to many variables, including the difficulties in determining a precise CH_4_ phenotype and the absence of cost-effective techniques that can reliably generate individual CH_4_ recordings for a significant number of animals. In order to include information on CH_4_ production in animal breeding programmes, it is necessary to have precise and cost-effective phenotypes, as well as enhanced techniques for accurately quantifying CH_4_ production [25].

Pszczola et al. [98] discovered that the heritability level of estimated methane emission in dairy cows varied throughout lactation. It began at 0.23, rose to 0.3 at 212 DIM, and concluded at 0.27. The mean heritability was 0.27, suggesting that estimated methane emission is a heritable characteristic and its magnitude varies throughout lactation [98]. A 2018 study conducted in Denmark revealed that the heritability of CH_4_ emission in Holstein cows was consistent with earlier findings, with a value of 0.25. The analysis, involving 2990 cows, showed that the selection index calculations indicated that residual methane had the most potential for incorporation into the breeding objective, in comparison to methane production (MeP; CH_4_ (g/d)), methane yield (MeY; CH_4_ (g/d)/DMI (kg/d)), and methane intensity (MeI; CH_4_ (g/d)/ECM (kg/d)) [25]. This is because residual methane enables the selection of animals with low methane emissions without affecting other economically significant features. Adding residual feed intake to the breeding objective might lead to a further decrease in methane emissions, since there is a moderate association between residual feed intake and methane levels, resulting in a positive and beneficial reaction. Introducing a negative economic valuation for methane might effectively decrease methane emissions while also allowing for an increase in milk production [96].

## 4. Methane Emissions and Heat Stress in Dairy Cow

The interaction between climate change and dairy farming is a complex and multi-dimensional phenomenon, with consequences that affect both local and global scales. On a global scale, the dairy business encounters difficulties such as increasing temperatures that cause heat stress in cattle, resulting in negative effects on their health, milk output, and fertility [99].

Heat stress (HS) tolerance is a crucial factor for dairy cattle who produce a high amount of milk, regardless of whether they are in tropical, sub-tropical, or temperate climates [100]. Even in moderate climate zones such as Central Europe, estimates for future climatic conditions, particularly during the summer months, suggest an increase in the frequency of heat waves and droughts. In relation to the impact on dairy cattle, who suffer from heat stress when exposed to elevated temperatures, the number of days leading to heat stress has already increased in various places over the previous few decades. Changes in physiological markers, such as body temperature or respiration rate, provide insights into immediate reactions to hot conditions, whereas impacts on animal behaviour and performance, particularly daily milk production, become apparent following prolonged exposure to heat [101]. Exposure to high environmental temperatures, humidity, heat waves, and sun radiation has been proven to raise the heat load in animals. This leads to negative effects on cattle homeostasis, thermoregulation, and normal physiology. Animals adapt to heat stress by modifying their normal physiology, behaviour, and metabolism, mostly through systemic neuronal and endocrine connections [102].

Reducing stress is an additional approach to enhance animal welfare and minimise the environmental consequences of dairy farming, in addition to promoting health improvement. Stress is a state of disrupted balance in the body’s internal equilibrium caused by internal events or external stimuli, typically resulting in elevated levels of cortisol. Animal’s suffering can arise not only from physical harm, but also from other stressors such as heat, social factors including isolation from others, or inadequate living space. Stress can elevate the metabolic rate and energy expenditure of cows, leading to higher levels of CH_4_ and N_2_O emissions. Furthermore, stress impacts factors that determine the animals’ capacity to emit, such as their feed conversion ratio and residual feed intake [8]. These factors contribute to the overall environmental impact of animal agriculture. In addition, stressed animals may exhibit altered behaviour and reduced immune function, making them more susceptible to diseases. This can lead to an increased use of antibiotics and other veterinary interventions, further contributing to the environmental footprint. Therefore, addressing animal stress and promoting their welfare not only improves their quality of life but also helps mitigate the environmental consequences of livestock production.

Ruminant productivity is significantly impeded by unfavourable climatic circumstances, particularly HS. HS decreases the amount of dry matter intake, which in turn, impairs the metabolism of energy and protein, resulting in higher levels of metabolic disorder, mineral imbalance, and several other health issues [103]. The temperature–humidity index (THI) is frequently used to assess thermal comfort in dairy cows. When the THI reaches 71, it indicates the critical upper limit at which heat stress begins. Conception rate decreases by up to 39% and significant economic losses occur when THI levels exceed 73 [8]. According to Lanzoni et al. [104], the summer season exhibited a substantial increase in THI, a decrease in DMI, and a decrease in emission intensities of Mediterranean buffaloes in contrast to the winter period [104]. The findings of a study in Italy showed that between day 1 and day 10, there was a decrease in the yield of CH_4_, and between day 10 and day 20, there was an increase. The interconnections among digestibility, passage rate, and DMI are intricate and will govern the quantity of intestinal CH_4_ generated [105]. Studies have shown that prolonged exposure to elevated temperatures may result in an elevation in ruminal temperature, which in turn, causes a decrease of 5% in ruminal pH [106]. In addition, these processes reduce the activity of microorganisms that digest cellulose in the rumen and alter its composition by increasing the presence of lactate-producing species and decreasing the presence of acetate-producing species in the population. This has the potential to impact milk production, as well as the movement of the rumen and the fermentation process that generates heat during feed digestion. An elevation in temperature and relative humidity leads to a reduction in the consumption of dry matter by animals and their rumination [62,105]. The inverse correlation between ambient heat and feed intake is amplified in the presence of large milk production. A decrease in the amount of food consumed leads to a drop in the heat produced by the organism, and this decrease is necessary to maintain a balance in the thermal load. Hence, it is evident that dairy animals with high productivity are more susceptible to heat stress [101]. Additional study is required to provide a comprehensive assessment of the direct impact of heat reduction methods on overall emissions in dairy farming [8].

## 5. Milk Production and Composition

The milk production per individual cow has consistently risen in numerous countries during the past few decades. Furthermore, this approach is frequently seen as a crucial tactic for reducing GHG emissions per kilogramme of milk generated, in addition to its potential economic benefits [107]. A study found that increasing milk production from 3200 to 7000 L per year per cow and using 750 kg of grain resulted in an increase in methane production from 379 to 460 g/day per cow. However, when considering methane production per litre of milk, it decreased from 35 to 21 g/L, indicating a 40% improvement in efficiency [108]. The enteric methane emissions and methane intensity (g CH_4_/kg milk) of the Australian dairy industry from 1980 to 2030 were analysed and estimated in a similar study. The findings of this research indicated that Australian dairy cows produced an average of 2889 kg of milk annually in 1980. By 2010, this figure had risen by 64%, to 5654 kg of milk annually. In 1980, it was estimated that the average methane intensity from dairy heifers in Australia was 33 g methane/kg milk (9.8 t CO_2_e/t MS). By 2010, this value had decreased to 20.2 g methane/kg milk (6.0 t CO_2_e/t MS) [109]. Figure 5 illustrates the correlation between milk amount and methane output per litre of milk.

Dairy cows exhibit varying levels of methane emissions across different physiological phases and over their lifetimes. A scientific study found that as lactation days increased, there was a decline in the DMI of dairy cows and a decrease in MY. Similarly, methane generation also followed this pattern, showing a strong positive link with these numbers. Irrespective of DMI, cows in the early lactation period exhibited the maximum methane output. The elevated methane generation observed during early lactation can be attributed to the heightened milk synthesis and the mobilisation of resources from somatic tissues. These factors have the potential to disrupt the energy balance of nursing cows. The rise in nutritional demand led to a concurrent increase in enteric methane emissions [23]. During the onset of lactation, cows experience a state of negative energy balance. Milk production relies on bodily energy, rather than energy derived from DMI. Thus, during the initial stage of lactation, there may be a negative association between MY and CH_4_ production due to the absence of CH_4_ generation from the energy utilised by the body. In the later stages of lactation, the energy required for milk production comes from the intake of dry matter, which leads to the creation of methane CH_4_. This results in a direct link between MY and CH_4_ production [110].

According to the study conducted by M. J. Bell et al. [111], they discovered notable impacts on CH_4_ emissions based on factors such as the week of lactation, daily MY, farm, and the interaction between daily MY and farm. CH_4_ emissions showed an initial increase during the first 20 weeks of lactation, followed by a period of stability from week 21 to week 50. The emissions ranged from 2.2 mg/L at week 1 to 3.2 mg/L at week 50. The impact of the lactation week on CH_4_ emissions can be attributed to variations in the quantity and composition of the consumed feed [111]. The type of the diet and the amount of feed consumed are significant elements that influence the variation in methane emissions from ruminant animals in their digestive system [112]. CH_4_ emissions are directly correlated with DMI and inversely correlated with the proportion of concentrates in the diet. Typically, DM intake is lower and contains a higher proportion of concentrate feed during the early lactation stage compared to later stages of lactation [111].

The quest for alternate measures of enteric methane emissions has broadened to include substances found in living organisms, including fatty acids (FAs) in milk [113]. Milk FAs show enormous potential as a methane proxy due to their direct correlation with microbial digestion in the rumen. The breakdown of carbohydrates in the rumen results in the generation of H_2_, and the process of methanogenesis is the necessary mechanism for eliminating this H_2_ [114].

Research findings indicate that there is a little positive relationship between the concentration of milk fat and methane output (with a slope of 0.067 g fat/g CH_4_ and a correlation coefficient of 0.281). Additionally, there is a strong positive correlation between the content of milk fat and methane intensity. The milk protein concentration exhibited a modest link with methane intensity, characterised by a slope of 0.030 g protein/g CH_4_ and a correlation coefficient of 0.375. The MY and milk fat concentration showed a moderate negative link, with a correlation coefficient (r) of −0.488 [115]. These findings suggest that an increase in the ratio of acetate to propionate in the rumen can lead to higher milk fat concentrations and methane emissions. This could be attributed to the fact that acetate is a precursor for milk fat synthesis, while propionate is utilised for energy production. Therefore, a higher proportion of acetate relative to propionate may result in an increased milk fat concentration. However, it also leads to higher methane production, as acetate is a major source of methane in the rumen.

## 6. Innovative Technologies Registered Physiological Parameters—Smart Dairy

Enhancing technological advancements at the agricultural level is a vital approach towards achieving enhanced environmental sustainability. Predictions indicate that the implementation of smart farming technologies will enhance the sustainability of the agricultural sector [116]. The development of accurate, low-cost, and simple proxies for CH_4_ production in individual animals may allow farms to apply low-CH_4_ management and breeding strategies [117]. Additionally, smart farming technologies will enable farmers to make informed decisions based on accurate and timely information, improving the overall profitability and resilience of the agricultural sector. One method to decrease emissions per unit of product in dairy farming is to enhance the MY per cow. This can be accomplished by implementing more efficient technologies, such as automatic milking systems.

Although there have been significant advancements in nutrition, there are still various other elements connected to animal physiology that may impact their bioenergetic efficiency and a decrease in GHG emissions. These factors require further investigation to gain a better understanding. There are data suggesting that the physiological and behavioural reactions to stress in dairy cows may be linked to increased generation of intestinal CH_4_ and decreased output [118].

The literature universally acknowledges that the implementation of smart farming, also known as precision farming, has beneficial impacts on the agricultural system. Furthermore, sustainable intensification of the agricultural sector is identified as a paramount challenge that needs to be addressed soon [119]. Precision livestock farming (PLF), a contemporary agricultural method, utilises innovative technologies and sensors that measure bio-signals like the Global Positioning System (GPS), accelerometers, temperature sensors, biosensors, and data analytics to drive this transition [6,120,121]. The real-time monitoring capabilities of PLF enable precise management of multiple elements of dairy farming. PLF plays a vital role in reducing methane and ammonia emissions, which are significant factors in climate change, by carefully monitoring and analysing data on animal health, feed intake, and waste management. This method allows farmers to maximise the nutritional content and ease of digestion of animal feed, resulting in decreased emissions of methane from the digestive system and enhanced overall efficiency of the farm [120]. Precision feeding is the most notable technology in PLF for mitigating GHG and ammonia emissions. Precision livestock feeding seeks to accurately align nutrition provision with the specific nutrient needs of individual animals, utilising real-time input from sensors [121].

The ability to perpetually and non-invasively monitor the activity and vital signs of farm animals in real time is gaining importance in the agriculture industry. This functionality enables consumers to have confidence that the animal is in good health and receiving proper care. Additionally, it promotes efficient management of the animal and its habitat, decreases unintended by-products from livestock farming, and enhances the productivity of farm inputs and resources [122]. The emergence of Big Data and Artificial Intelligence (AI) has completely transformed the dairy farming industry, bringing about increased productivity, higher environmental friendliness, and innovative management techniques. These technologies tackle principal issues in dairy production, such as maximising resource utilisation, enhancing herd well-being, and establishing sustainability benchmarks [123]. AI technologies enhance herd management and optimise resource allocation. The utilisation of AI-driven predictive analytics in herd health brings about a transformative impact on animal care by proactively anticipating health problems, hence improving preventative treatment [122]. The emergence of AI technology has presented novel prospects for more accurate assessment of methane emissions in the livestock business [15].

### 6.1. Body Weight

To maintain important levels of productivity without compromising their metabolic state, cows must possess the ability to rapidly recover from the period of negative energy balance they experience during early lactation. Hence, it is preferable to choose cows with a greater genetic advantage for the body condition score (BCS) [25]. The BCS of a dairy cow is a measure of its adiposity, or the amount of body fat it has [124]. The BCS is an essential metric utilised in the evaluation of cattle welfare, carrying substantial consequences for health, productivity, and reproductive achievement [61]. Overall, the relationship between body weight and CH_4_ production is complex and influenced by several factors such as lactation stage and feed intake. While there is a positive correlation between body weight and CH_4_ production during periods of increasing feed intake, this correlation becomes negative during periods of decreasing feed intake, such as at the beginning and end of lactation. Therefore, it is important to consider these factors when studying the relationship between body weight and CH_4_ production in livestock [110].

The findings of Zetouni et al. [25] indicate a positive correlation between reproductive features and CH_4_ production, which aligns with the findings of other researchers regarding BCS. Improved BCS in dairy cattle is associated with enhanced fertility, suggesting that cows with fewer reproductive complications would emit lower levels of CH_4_ [25]. Research shows that body weight is commonly used as a key indicator for enteric methane production in dairy and beef cattle. In general, larger animals have elevated maintenance needs, resulting in increased food consumption and CH_4_ production [114]. As the daily feed intake rises, methane generation often increases. Currently, CH_4_ is a by-product of rumen fermentation in the digestive process of animals. According to a study, cattle create around seven to nine times more CH_4_ than sheep and goats [125].

### 6.2. Rumination

The metabolic status and illness state of dairy cattle around the time of parturition relate to the amount of time that they spend ruminating. Therefore, rumination monitoring may be useful for obtaining information on the health status of animals in a short amount of time during a crucial moment such as the transition phase. Early identification of clinical and subclinical disease through rumination monitoring enables farmers to promptly commence treatment, thereby reducing the expenses linked to managing chronic cases and mitigating more substantial output losses [126].

The presence of large levels of fibre in the diet leads to an increase in rumination activity, including its frequency, intensity, and length. Rumination serves a dual function in regulating intake and digestion [31]. Various environmental factors experienced by ruminants can influence their rumination behaviour. For instance, the presence of stressful occurrences such as infections, metabolic diseases, and heat stress may have an adverse impact on rumination, influencing both animal wellbeing and production. Furthermore, the rumination process can be influenced by several dietary and nutritional parameters, including the digestibility of the feed, the intake of fibre, and the quality of the forage. These factors have the potential to either enhance or diminish rumination [127].

Initially, rumination stimulates the formation of saliva, which helps to counteract the fall in pH in the rumen [127]. This promotes the process of acetogenic fermentation, leading to an increase in the synthesis of milk fat. However, it also results in the generation of more enteric CH_4_. Furthermore, the act of chewing during rumination serves to decrease the size of the food particles and enhance the surface area available for fast fermentation by the microorganisms present in the rumen. Therefore, rumination directly influences the rate at which food is digested, which in turn, can affect the regulation of both feed intake and methane production [31]. For this reason, the measurement of rumination time has been proposed as an alternative to estimate the amount of methane emitted [128,129]. Although the respiration chamber is considered the benchmark for CH_4_ recording, it is a costly and labour-intensive approach. Estimating and monitoring rumination time could serve as an alternative method for predicting intestinal methane excretion. The presence of elevated levels of fibre in the animal’s diet leads to an increase in rumination. Furthermore, the act of ruminating stimulates the formation of saliva, which helps to regulate the pH of the rumen. The duration of rumination can impact the size of food particles by impacting the chewing process. This, in turn, can influence the microbial fermentation process, which has an impact on the movement of food through the digestive system. This can affect the regulation of feed intake by influencing the feeling of fullness. Consequently, it could affect the production of methane gas. Commercial herds can already monitor rumination time on a large scale using a non-invasive sensor-based technology. This system captures the sounds made by the cow’s jaw or ear movements to determine rumination activity [128].

Prior research has primarily concentrated on the eating behaviour or rumination, as these activities directly contribute to the fermentation process, which results in the production of methane [130]. The study found a negative connection between the duration of rumination and both methane concentrations (−0.24 ± 0.38) and methane production (−0.43 ± 0.35). This indicates that animals that spend more time ruminating tend to release less CH_4_ when exhaling. Several factors that impact CH_4_ generation can likewise influence the duration of rumination. Feeding rations that contain a significant amount of forage neutral detergent fibre increases the activity of rumination and enhances the formation of acetate in the rumen, resulting in greater methane generation. Rumination stimulates the formation of saliva, which helps to regulate the acidity of the rumen, promotes the creation of acetate, and decreases the generation of methane [131]. According to the findings in 2022, low rumination cows exhibited the highest daily CH_4_ production, emitting 1.8% more CH_4_ than medium-rumination cows and 4.2% more than low-rumination cows. Compared to cows in the median-rumination group, those in the high-rumination group produced 2.9 percent less CH_4_ per milk unit; conversely, cows in the low-rumination group produced 4.6 percent less CH_4_ [132]. Nevertheless, several investigations have demonstrated that rumination does not have an impact on the levels of CH_4_ or CO_2_ [133].

### 6.3. Chewing Rates

Numerous early studies established the groundwork for our current comprehension of the mechanisms underlying chewing, the physiological significance of chewing in bovine health, and the ways in which dietary attributes influence chewing behaviour. Recent publications in the profusion of the literature offer fresh perspectives on the feeding and ruminating behaviour of dairy cows [134]. Chewing when eating and ruminating is crucial for digestion and the transportation of feed through the dairy cow’s gastrointestinal tract [135]. The literature is replete with evidence that encouraging chewing in dairy cows increases salivary secretion, thereby mitigating the risk of acidosis [134]. For example, dairy cows subjected to HS exhibited reduced feed intake and chewing activity on a regular basis. To maintain rumen health, dairy cows necessitate prolonged chewing activities. As a consequence, reduced chewing activities may lead to a decrease in the alkalinity of the saliva in the rumen, thereby compromising the health of the rumen and ultimately affecting the productivity and performance of dairy cows [136]. According to research, there is a direct correlation between the rate of chewing and methane production within the range of 68–120 chews per minute. The Total Mixed Ration (TMR)-fed group exhibited the highest chewing rates, averaging 100 chews per minute, while both grass-fed groups had lower rates, averaging 78 chews per minute. The higher digestibility of TMR compared to grass, along with its smaller particle size and increased surface area, is the most probable reason. Hence, rumen microorganisms exhibit a higher rate of digestion and methane production while processing TMR as opposed to grass [30]. Overall, the findings from this research highlight the importance of considering feed composition and chewing rate when studying methane emissions in ruminant animals.

### 6.4. Temperament

The level of stress response in cattle can have a substantial impact on the variation in feed efficiency. Cattle that are more excitable tend to experience higher levels of stress and are less efficient in converting feed into desired outcomes. Greater feed efficiency is indeed linked to a calmer temperament and a reduced physiological stress response, specifically in the form of lower cortisol levels. While there is debate surrounding the direct impact of residual feed intake (RFI) on CH_4_ emissions, it is commonly acknowledged that choosing more productive animals at the individual level will lower emissions intensity (measured in grammes of CH_4_ per unit of product), hence reducing GHG emissions [137]. The results from P. Llonch et al. [137] show that there is a negative relationship between the number of times a cow eats and the amount of CH_4_ it releases. This means that if dominant cows ate more often and for longer periods of time, there would be less competition at the feeder, which would improve feed efficiency and lower CH_4_ emissions [137].

According to Marçal-Pedroza et al.’s [118] research, cows that kicked the milking cluster off more often spent 25.24% less net energy on lactation and exhaled 36.77% more enteric CH_4_/kg of milk. Cows that ruminate more at the milking parlour spent 50.00% more metabolizable energy, 57.93% more net energy, and 37.10% less CH_4_/kg of milk [118]. These findings suggest that the behaviour of cows during milking can have a significant impact on their energy expenditure and methane emissions. This can change how metabolism works by affecting the hypothalamic–pituitary–adrenal (HPA) axis. This can make the HPA axis more active, which releases glucocorticoids, which are hormones that help with the stress response. Cattle with reactive temperaments, characterised by high flight speed and crush scores, exhibited a longer and more intense activation of the HPA axis and sympathy–adrenomedullary system in response to stress. Both axes play a role in regulating catabolism, maintaining energetic balance, managing energy levels, and storing energy in the body [118]. Additionally, these findings suggest that cattle with reactive temperaments may have a higher risk of energy imbalances and difficulties in energy storage within their bodies.

The herd of Belgian Blue cattle was assessed individually in the field, and their behaviour was observed, revealing a distinct daily pattern of activity characterised by heightened grazing activity after dawn and before nightfall. Nevertheless, no discernible methane emission pattern could be linked to it, since the daily fluctuation in emissions was smaller than the level of accuracy in the measurements [138]. It is important to note that the relationship between temperament or behaviour and methane production is not entirely straightforward, and several factors can interact to produce different outcomes. Still, additional comprehensive research is necessary in the future, given their scarcity and the abundant research prospects they provide.

### 6.5. Infrared Thermography

Infrared thermography (IRT) is a non-invasive approach that has been researched as a tool for detecting various physiological and pathological processes associated with alterations in body temperature in real time [139,140]. IRT now offers the ability to measure not only the body temperature of cattle, but also to perform other types of detections [141].

Heat generation can serve as a criterion for designing dietary approaches to enhance feed efficiency, as it is an essential factor in measuring the rate of metabolic energy conversion. The assessment of methane production in ruminants aligns with this reasoning as it signifies energy dissipation, with its much-debated significance in environmental concerns [142]. According to Yuri R. Montanholi et al. [143], infrared thermography can effectively be used to evaluate heat and methane generation by analysing the temperature of the feet and the temperature difference between the left and right flanks, respectively. This method is additionally beneficial for evaluating physiological reactions to the processes of milking and feeding. Therefore, the disparity in temperature between these two body sites will indicate the variations in rumen temperature. The decrease in the temperature disparity between the sides of the body following the initial meal is indicative of the cooling impact of feeding on the rumen. Conversely, the subsequent rise in the temperature disparity signifies the heat generated during fermentation in the rumen [143]. Anne R. Guadagnin et al. [142] discovered a significant association between methane release in Holstein cows and the thermography of their eyes, which was captured 5 or 6 h after feeding [142]. The highest levels of methane production were found within 6 h after feeding, which corresponds to the time when there is a greater abundance of H_2_ available for the methanogens [143]. The correlation between IRT and methane release can be elucidated by the endothermic and exothermic rumen activities. The process of methanogenesis from the breakdown of acetate is characterised by being an endothermic reaction, while autotrophic methanogenesis, which occurs from the reaction between CO_2_ and H_2_, has exothermic properties. The process’s proportionality relies on the rumen environment. IRT is employed to assess the internal conditions of the rumen by measuring its temperature during the response, which in turn, affects the overall body temperature [144].

## 7. Breeding Programmes

For many decades, the primary focus of animal breeding has been on increasing production efficiency. Presently, dairy production systems employ a variety of reproductive technologies, including artificial insemination, embryo transfer, and sexed semen. Additionally, a selection of hormone therapies is accessible to assist in achieving fertility or to synchronise oestrus and/or ovulation, thereby facilitating calving [145]. Recent research has demonstrated the feasibility of choosing cows that release lower levels of CH_4_ [146]. Genetic selection of low-CH_4_-emitting heifers represents a prospective approach to mitigating the dairy industry’s environmental impact. In contrast to the extensively researched domains of management and nutrition, genetics possesses the advantage that the consequences of selection are cumulative and enduring [22]. Even though it has a less significant effect in the short run [147]. The objective of animal breeding is to manipulate the ruminal microbiota through selective processes to produce a more efficient microbial composition. This composition should result in reduced utilisation of natural resources and lower methane emissions while maintaining good health and productivity [148]. Therefore, choosing animals that generate low levels of CH_4_ could result in substantial decreases in emissions over a few generations [149]. By selecting animals with a more efficient microbial composition, farmers can potentially reduce the environmental impact of methane emissions while still maintaining elevated levels of animal health and productivity.

Recent genetic investigations pertaining to dairy cattle have unveiled that although non-genetic factors such as feed, management, and environmental influences account for most of the variability observed in CH_4_ production, genetic variance, which constitutes the genetic component, is also present in CH_4_ production [150]. Extensive research has been conducted and is ongoing to find technical solutions that can effectively minimise enteric CH_4_ emissions [22]. The practice of animal breeding has consistently demonstrated its ability to enhance productivity and decrease vulnerability to diseases. Moreover, it holds the potential to aid in the reduction of methane emissions originating from cattle [151]. Breeding cows with lower methane emissions will contribute to achieving the targets established at the 2015 Paris COP meeting [7,152]. In their study, J. Lassen et al. [129] also examined the genetic correlations between CH_4_ emissions and other traits that are important for breeding purposes, such as reproduction and health. They found that selecting for reduced CH_4_ emissions has only a small impact on these traits. However, further analysis using larger datasets is required to confirm or refute the genetic correlation patterns of other traits [129].

Several studies have been conducted to examine the impact of different cow breeds on methane emissions. According to D. W. Olijhoek et al. [153], the reaction to increased concentrate feeding varied by breed, with Holstein cows having a lower methane yield (methane per kg of DMI) than Jersey cows when the concentrate fraction in their diet was raised. When increased levels of dietary concentrate were provided to Holstein cows, the rumen acetate–propionate ratio decreased more than that of Jersey cows. Methane intensity (methane per kilogramme of energy-corrected milk (ECM)) was identical in Holstein and Jersey dairy cows [153]. Additional findings from researchers have demonstrated, that under conditions of optimal nutrition, purebred Gyr and Holstein X Gyr heifers exhibit superior feed efficiency in comparison to Holstein. Holstein heifers that are fed at a medium feeding level exhibit superior feed efficiency and reduced CH_4_ emissions in comparison to Gyr heifers [154]. These findings suggest that diet composition and breed can have a significant impact on rumen fermentation and methane emissions in dairy cows. The decrease in rumen acetate–propionate ratio in Holstein cows indicates a more efficient utilisation of feed energy, potentially leading to improved milk production. However, the similar methane intensity in both Holstein and Jersey cows suggests that breed may not play a significant role in methane emissions. This highlights the importance of optimising nutrition and feed management practices to enhance feed efficiency and reduce GHG emissions in dairy farming.

Breeding systems that are meticulously designed have the potential to generate animals that possess inherent low methane emissions. Trait interdependence is a critical consideration when devising a breeding strategy, as modifications to one trait can lead to alterations in other traits, consequently impacting producer profits [155].

## 8. Conclusions

Global dairy systems face mounting pressure to mitigate GHG emissions [3]. Enhanced animal well-being and decreased rates of death and illness are anticipated to lead to higher levels of animal output, hence reducing GHG emissions per unit of product. To the best of our knowledge, limited attention has been paid to measuring GHG emission intensity and its effects on dairy cow health, including both subclinical and clinical disorders. When it comes to diseases in dairy cows, the first signs are usually a decrease in both feed intake and MY. Proposing the adoption of emission-reducing strategies and the preservation and upkeep of optimal animal welfare as viable methods to enhance the sustainability of contemporary dairy production [8]. Without healthy animals, the entire industry would be at risk of collapse. They not only contribute to the production of high-quality meat, milk, and eggs but also play a crucial role in maintaining the balance of ecosystems. Furthermore, healthy animals are essential for genetic diversity, which allows for the development of disease-resistant breeds and contributes to the long-term sustainability of animal agriculture. Ultimately, prioritising the health and welfare of animals is not only ethical but also vital for the future of the industry.

Additional data on animal health before animal mortality are needed to increase the accuracy of the GHG emission intensity estimate. By analysing animal health data prior to mortality, a more precise estimation of GHG emission intensity can be achieved. This would enable dairy farmers to have a clearer understanding of the environmental impact of their operations and make informed decisions regarding emission-reducing strategies. Furthermore, prioritising optimal animal welfare not only contributes to the sustainability of dairy production but also ensures the well-being of the cows, which in turn, can lead to improved overall productivity and reduced disease prevalence. Thus, a comprehensive approach that considers both environmental and animal health factors is crucial to promoting a sustainable and efficient dairy industry. In addition, implementing technologies such as precision farming and data analytics can further enhance the efficiency and sustainability of dairy operations.

## 9. Future Implications and Progressive Pathways

With the increasing stringency of environmental rules, the dairy business is compelled to persistently innovate and adapt. The dairy farming sector is currently facing a crucial moment where it is necessary to decrease GHG and ammonia emissions. This is not only an environmental necessity but also a legal obligation. The adoption of innovative techniques and adherence to developing policies will have a significant impact on the industry’s future. The interaction between laws, technology, and farming methods will determine the direction towards a dairy industry that is more sustainable and environmentally conscious [99]. Additionally, the use of precision farming techniques and advanced monitoring systems can help optimise resource utilisation, thereby minimising ammonia emissions. It is crucial for dairy farmers to stay updated on evolving regulations and actively participate in research and development to ensure compliance with environmental standards. The industry’s commitment to a sustainable and eco-friendly approach will not only benefit the environment but also enhance its reputation and ensure long-term profitability.

Insufficient research has been conducted to investigate the health of dairy cows and their contribution to GHG emissions. Hence, it is imperative to persist in scientific investigation in this domain to mitigate the release of GHG into the environment. To effectively incorporate pathogens into climate models and mitigation plans, it is crucial to get a comprehensive understanding of the broader impact of infectious illnesses on ecosystems, beyond their direct effects on host health and the global food supply [73]. Analysing the effects of diseases via a One Health perspective and applying the same framework to evaluate present and future management methods and production systems to enhance animal, human, and ecosystem health should enhance sustainability. However, for this to be successful, it is crucial to properly explain the significance of enhancing cow health as a method to promote sustainability to all parties involved [43].

By emphasising the interdependence of animal, human, and ecosystem health, it becomes evident that investing in cow health not only benefits the animals but also has wider implications for the environment and society. This comprehensive understanding of the broader impact of infectious illnesses on ecosystems can guide the development of effective management methods and production systems, leading to a more sustainable future.

## Figures and Tables

**Figure 1 animals-14-00829-f001:**
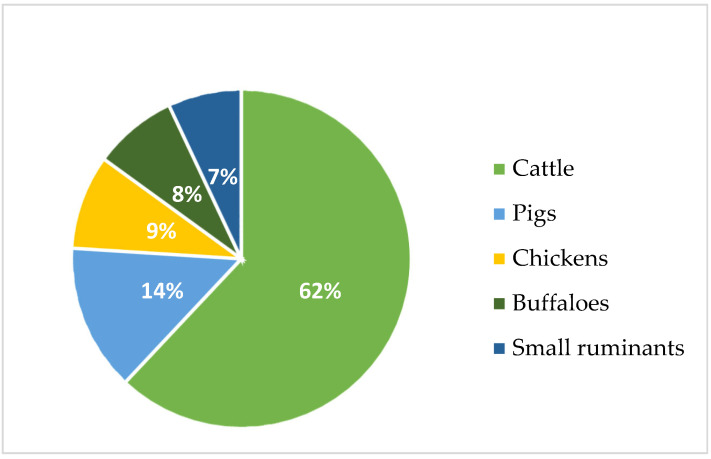
Animal category proportions on enteric methane emissions (FAO 2023) (illustration generated using BioRender.com).

**Figure 2 animals-14-00829-f002:**
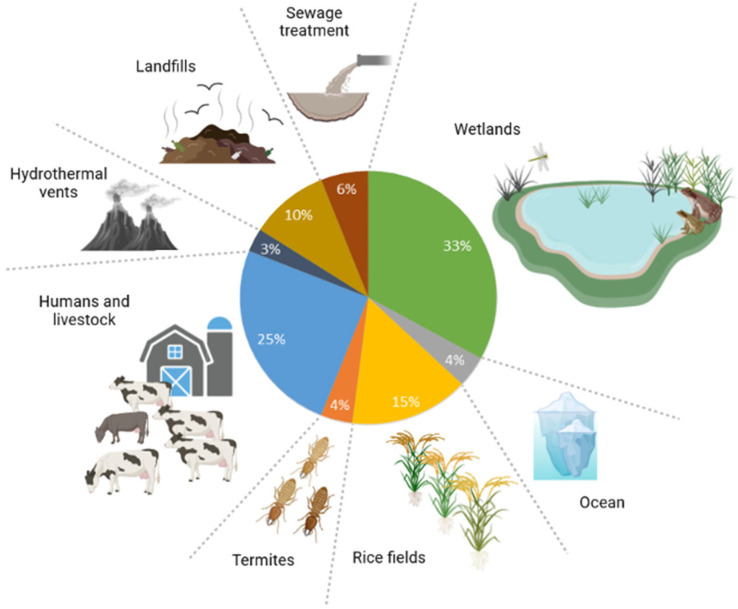
Biological entities that release methane into the atmosphere (illustration generated using BioRender.com).

**Figure 3 animals-14-00829-f003:**
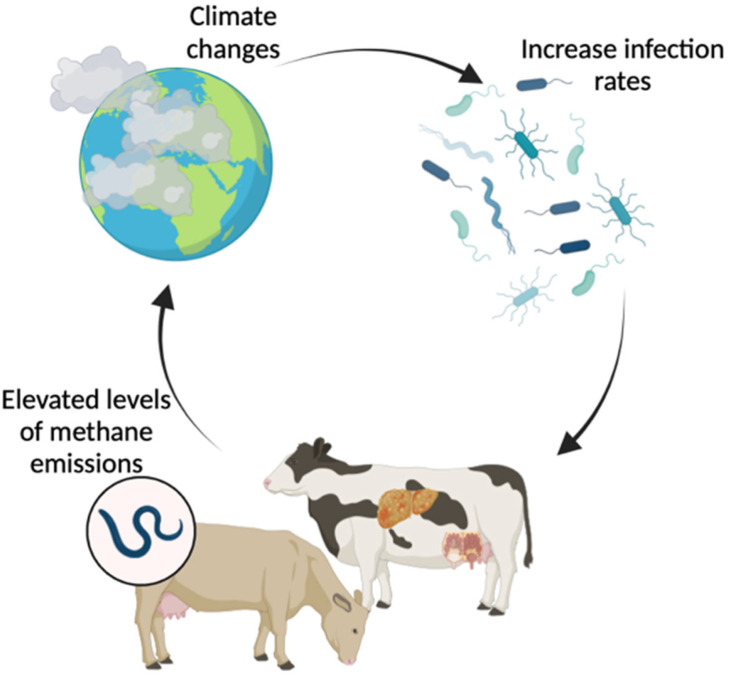
The relationship between climate change and cattle infections (illustration generated using BioRender.com).

**Figure 4 animals-14-00829-f004:**
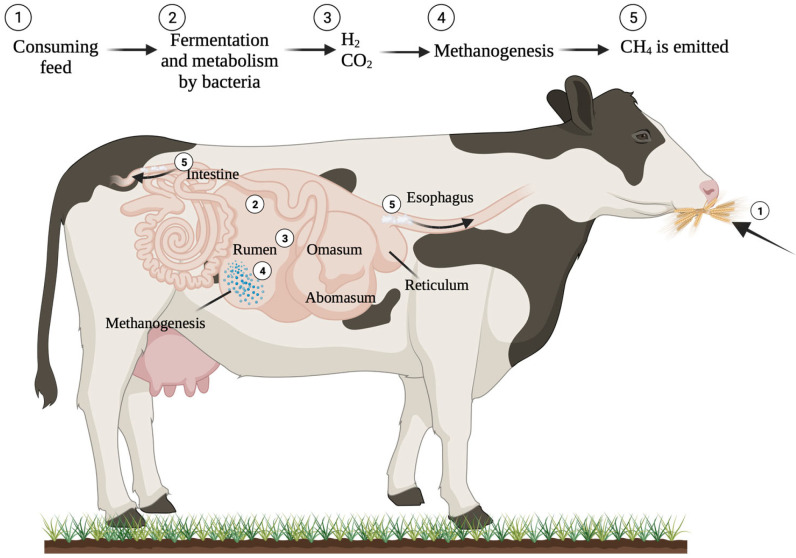
Mechanisms involved in the generation of methane in ruminant (illustration generated using BioRender.com).

**Figure 5 animals-14-00829-f005:**
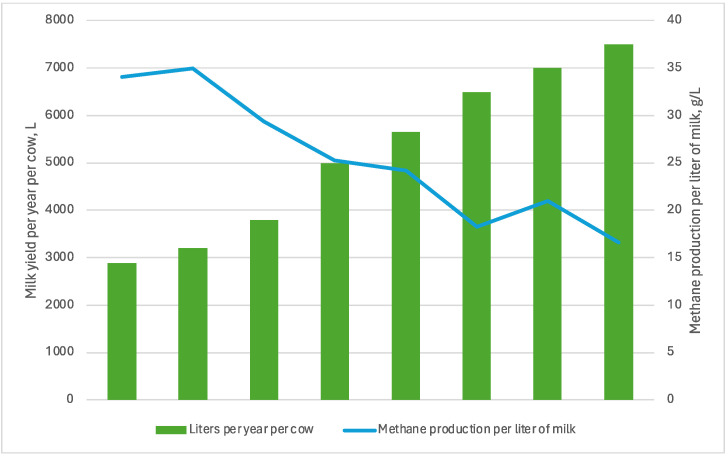
Methane production and milk yield per cow year.

**Table 1 animals-14-00829-t001:** The effects of cattle diseases on economic expenses, milk production, and emissions of GHG.

Disease	Economic Cost	Milk Yield	^c^ Possible Impact on GHG per Unit of Milk or Meat Produce	References
Bovine viral diarrhoea virus	^b^ Up to EUR 294.48 per cow (with an average of €54.34 per cow)	Reduced	Moderate	[43]
Infectious bovine rhinotracheitis	^b^ EUR 233.71 per sub-clinically infected cow	Reduced	High	[43]
Bovine respiratory disease	EUR 21.77 per case	N/A	Low	[43,44]
Johne’s disease	^b^ EUR 30.38 per dairy cow and EUR 19.87 per beef cow	Reduced	High	[43]
Mastitis	Chronic EUR 118. Quartile range EUR 106–132 per case, clinical EUR 240/lactating cow per year	Reduced	Moderate	[43,45,46]
Lameness	From EUR 214.51 to 992.10 per case	Reduced	Moderate	[43,47,48]
Subclinical ketosis	EUR 179.37 per calving cow	Reduced	Low	[49,50,51,52]
Clinical ketosis	single clinical ketosis case averaged EUR 709	Reduced	N/A	[42,53,54]
SARA	^a^ of EUR 368.59 to EUR 437.71 lost income per cow per year	Reduced	N/A	[55]

^a^ USD 1 = approximately EUR 0.92 on 27 February 2024; ^b^ GBP 1 = approximately EUR 1.17 on 27 February 2024; ^c^ low = <4%, moderate = 4–8%, high = >8%.

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
