# Peer review of "Relationship between Dairy Cow Health and Intensity of Greenhouse Gas Emissions"

_animals, 2024, doi:10.3390/ani14060829_

Round 1

Reviewer 1 Report

Comments and Suggestions for Authors

Author Response

Comments and Suggestions for Authors

Authors comments, corrections and answers

General comment

The objective of the paper is in accordance with the aim of the Journal.
The manuscript is interesting because it has gathered accurate information regarding animal health and its close relationship with greenhouse gas emissions. However, some inferences and hypotheses may need more support from the literature and must be clarified. Not to mention the fact that some contradiction has been found and needs to be clarified or stated from a point of view. Therefore, this manuscript needs some minor revisions before publishing.

Thank you for the kind remarks.

Simple Summary

L19: “…to have a more comprehensive comprehension of…” repetition, correct the sentence

This line was changed to: “This will allow dairy producers to have a better understanding of the environmental consequences of their operations and make well-informed choices regarding ways to reduce emissions.”

Abstract

L29: correct “The emissions..” for “These emissions...”

We corrected this sentece: “These emissions mostly originate from methane (CH4), nitrous oxide (N2O), and carbon dioxide (CO2).”

Chemical formulas that are written as N2O, CO2, CH4, H2 need to be in the correct format. Revise and make corrections throughout the whole manuscript.

We rewritten formulas N2O, CO2, CH4 and H2 in to: N2O, CO2, CH4 and H2 .

L45: use a better word instead of “cattle sector”

We corrected this sentece and used “livestock industry”: “Through the implementation of these measures, the livestock industry may enhance both animal well-being and mitigate the release of methane and nitrous oxide, thereby fostering environmental sustainability. In addition, advocating for sustainable farming methods and providing farmers with education on the significance of mitigating GHG emissions can bolster the industry's endeavours to tackle climate change and infectious illnesses.”

L53: add a in policymaking so it is policy-making

We corrected this sentece: “It aims to identify areas where research is lacking and to provide guidance for future scientific investigations, policy-making, and industry practices.”

L58: add a ; between the key words: methane emissions and cattle

We added: “… methane emissions; cattle; …”

Content

Table 1. The economic cost for each disease has not the same currency (pounds, euros) and not the same unit (sometimes is per cow, per year, to the industry, per case, etc). So it is difficult to understand or compare the economic impact of these diseases. If it is possible, try to make conversions and use the same unit.

Table 1 shows the corrected economic cost of each disease in euros.

United States Dollar 1 = 0.92 Euro (2024.02.27)

Pound sterling 1 =  1.17 Euro (2024.02.27)

In Table 1 we changed the unit (per case, per cow or per cow per year). Since comparing economic losses was not the objective in this instance, we have retained the adjusted units of measurement. These units of measurement were documented in the literature that was examined.

L344,345 Use the cursive type of letter for the scientific names Escherichia coli, Klebsiella pneumoniae, and Staphylococcus aureus, and add the word respectively at the end of the sentence.

We corrected this sentece: “United States resulted in an increase in antibiotic resistance of 4.2%, 2.2%, and 2.7% for the common pathogens Escherichia coli, Klebsiella pneumoniae, and Staphylococcus aureus respectively.”

L402 Add a comma between identified and examined

We corrected this sentece: “One of the limited number of studies identified, examined the effects of Fasciola hepatica in beef cattle.”

L403 Add a comma after cattle

We corrected this sentece: “In beef cattle, the observed 1.5% rise in GHG emissions intensity in 2022 attributed to Fasciola hepatica seems to be moderate.”

L482 Revise the whole sentence

We corrected this sentece: “Recent studies have demonstrated that dairy cattle have a moderate degree of heritability with regard to CH4characteristics.”

L558 Revise the sentence: “The findings of study I Italy showed that between..”

We revised the sentece: “The findings of study inItaly showed that between day 1 and day 10, there was a decrease in the yield of CH4, and between day 10 and day 20, there was an increase.”

L578-585 As a suggestion, take this paragraph and make a chart or table with a more explicit explanation of how different is to compare methane production using “gram of methane per cow” or “grams of methane per litre of milk.” More info and more data will be needed but as the authors have gathered accurate information regarding the whole topic this is not going to be an issue.

We created a graph depicting the relationship between milk quantity and methane emissions per litre of milk (Figure 5).

L700,701 Is there any scientific work showing any number for the relation of larger animals and methane production mentioned?

In this secion we added information: “As the daily feed intake rises, methane generation often increases. Currently, CH4 is a by-product of rumen fermentation in the digestive process of animals. Cattle create around 7 to 9 times more CH4 than sheep and goats.”

L733,734 and L750,751 These two sentences expressed a contradiction between the saliva and the methane production. The first one states that more saliva increases methane and the last one the opposite. State and define a clear point of view of this two opposite assumptions.

We have corrected these sentences and misunderstanding. We left the main opinion of scientists that rumination is related to the amount of methane released. As rumination increases, methane gas release decreases.

Reviewer 2 Report

Comments and Suggestions for Authors

Line 34 - What is the basis for using CH4/unit rather thsn CH4 grams or liters? Just a couple of comments would help. Globally, it is the total emissions that affect the atmosphere. 

Lines 64-87   reduction?me notation such as "cited by" woul be appropriatr. This check should be done throughout all of the paper.

Lines 112 and 115 - Duplicate sentences.

Line 129 - Reference for the 20%. Is this reduction per unit or total CH4? Clarify. Check for this throughout the rest of the paper.

Table 1- Is the GHG column per uint or total CH4. Add this to the column label or as a footnote. The table is good and very useful. Including the economic cost column is excellent.

Line 290 - Is this per unit or total? Needs to be defined in the text.

Line 468- You should put the reference in parenthesis after the et al. Do this throughout the paper. Same for line 476.

Line 558- study I Italy?  Reword. Should this  be in rather than I?

Overall, check for consistency in indicating per unit or total when indicating changes in CH4 and check whether cited material is directly from the  reference listed or if the text shoud reflect that it was cited in the reference listed.

Author Response

Comments and Suggestions for Authors

Authors comments, corrections and answers

Line 34 - What is the basis for using CH4/unit rather thsn CH4 grams or liters? Just a couple of comments would help. Globally, it is the total emissions that affect the atmosphere.

In this section we added infromation about CH4 unit: “This CH4 unit approach allows for a more accurate comparison of emissions across different animal production systems, considering variations in productivity. Expressing methane emissions per unit allows for easier comparison between different sources of emissions. Expressing emissions per unit (e.g., per cow) highlights the relative impact of these sources on the environment. By quantifying emissions on a per-unit basis, it becomes easier to identify high-emission sources and target mitigation efforts accordingly. Many environmental policies and regulations focus on reducing emissions per unit of activity or output. By focusing on emissions per unit, policymakers and producers can work together to implement practices that lower emissions without sacrificing productivity. Expressing methane emissions in this way aligns with policy goals aimed at curbing overall greenhouse gas emissions. While it's true that total emissions affect the atmosphere globally, breaking down emissions per unit helps to understand the specific contributions of different activities and sectors to overall greenhouse gas emissions.”

Lines 64-87   reduction?me notation such as "cited by" woul be appropriatr. This check should be done throughout all of the paper.

We appreciate your comment, but could you perhaps provide further clarification on what exactly you mean?

Thank you.

Lines 112 and 115 - Duplicate sentences.

We removed one sentence. “They then convert CO2 into CH4 through reduction and are the main producers of methane in deep marine sediments, termite hindguts, and the gastrointestinal tracts of humans and animals. Methanogens are the primary producers of methane in deep sea sediments, termite hindguts, and the gastrointestinal systems of humans and animals. Collectively, these sources account for one-third of the methane emissions produced by living organisms (Figure 2).”

Line 129 - Reference for the 20%. Is this reduction per unit or total CH4? Clarify. Check for this throughout the rest of the paper.

1.     In this reference is a total CH4. This sentence reference is “Beauchemin, K.A.; McGinn, S.M. Reducing Methane in Dairy and Beef Cattle Operations: What Is Feasible?”And we corrected this sentence: “A 20% reduction in total CH4 production could allow growing cattle to gain an additional 75 g/d of body weight and 1 L/d more milk yield (MY) from dairy cows”.

2.     We corrected this sentence: “Repeatedly infecting ewes with Teladorsagia circumcinctainfective larvae resulted in a 16% increase in total methane yield and a 4% increase in totalnitrous oxide yield per unit of dry matter intake. Similarly, per unit of digestible organic matter intake, there was a 46% increase in total methane yield and a 31% increase in total nitrous oxide yield.”

3.     We corrected this sentence: “Just recently, Fox et al. discovered that lambs infected with abomasal parasites exhibited a 33% increase in total CH4 output compared to uninfected animals.”

Table 1- Is the GHG column per uint or total CH4. Add this to the column label or as a footnote. The table is good and very useful. Including the economic cost column is excellent.

We corrected Table 1: “Possible impact on GHG per unit of milk”

Thank you for the kind remarks.

Line 290 - Is this per unit or total? Needs to be defined in the text.

We corrected this sentence: “Using real-time measured reticulorumen parameters, Antanaitis et al. discovered that dairy cows whose reticulorumen pH ranged from 6.22 to 6.42 had an average total methane emission increase of 46.18%.”

Line 468- You should put the reference in parenthesis after the et al. Do this throughout the paper. Same for line 476.

We put the reference in parenthesis after the et al. And we did it all on paper.

Line 558- study I Italy?  Reword. Should this  be in rather than I?

We revised the sentece: “The findings of study in Italy showed that between day 1 and day 10, there was a decrease in the yield of CH4, and between day 10 and day 20, there was an increase.”

Overall, check for consistency in indicating per unit or total when indicating changes in CH4 and check whether cited material is directly from the  reference listed or if the text shoud reflect that it was cited in the reference listed.

We appreciate the suggestions and commentary. The citations were thoroughly reviewed and rectified in the entire manuscript.
